# Misery Loves Complexity: Exploring Linguistic Complexity in the Context of Emotion Detection

**Pranaydeep Singh**[◇]**, Luna De Bruyne**[+,◇]**, Orphée De Clercq**[◇] **and Els Lefever**[◇]
[◇]LT3, Language and Translation Technology Team, Ghent University
`firstname.lastname@ugent.be`
[+]CLiPS, University of Antwerp
`firstname.lastname@uantwerpen.be`

## Abstract

Given the omnipresence of social media in our society, thoughts and opinions are being shared online in an unprecedented manner. This means that both positive and negative emotions can be equally and freely expressed. However, the negativity bias posits that human beings are inherently drawn to and more moved by negativity and, as a consequence, negative emotions get more traffic. Correspondingly, when writing about emotions this negativity bias could lead to expressions of negative emotions that are linguistically more complex. In this paper, we attempt to use readability and linguistic complexity metrics to better understand the manifestation of emotions on social media platforms like Reddit based on the widely-used GoEmotions dataset. We demonstrate that according to most metrics, negative emotions indeed tend to generate more complex text than positive emotions. In addition, we examine whether a higher complexity hampers the automatic identification of emotions. To answer this question, we fine-tuned three state-of-the-art transformers (BERT, RoBERTa, and SpanBERT) on the same emotion detection dataset. We demonstrate that these models often fail to predict emotions for the more complex texts. More advanced LLMs like RoBERTa and SpanBERT also fail to improve by significant margins on complex samples. This calls for a more nuanced interpretation of the emotion detection performance of transformer models. We make the automatically annotated data available for further research at hf.co/datasets/pranaydeeps/CAMEO.

## 1 Introduction

The *negativity bias*, coined by Kanouse et al. (1987), states that human beings are inherently drawn to, and impacted more by negative experiences, emotions, and interactions than by their positive counterparts. This cognitive bias may be more present than ever because of the evolution of social media which makes sharing thoughts and opinions

easier than ever before. Recent research by Rathje et al. (2021) has shown that publishing negative criticism or insults against political opposition is a highly effective way to acquire engagement on social media platforms. Similar research in the news domain shows that headlines with negative keywords generate more engagement and clicks than headlines with positive keywords (Robertson et al., 2023). While these studies explore the negativity bias from a consumer's perspective, in this work we explore the idea of a negativity bias from the perspective of the author and more specifically explore the correlations between negativity and complex language use.

The goal of this research is thus to further investigate the negativity bias in social media texts by exploring whether there is a link between emotion and text complexity. This overarching research goal will be broken down into two specific research questions. (1) Are negative emotions on social media more complex than positive emotions? (2) Does a text's linguistic complexity make it harder to model certain emotions using transformers, and do transformers that are better at understanding more complex text – such as SpanBERT (Joshi et al., 2019) – perform better than BERT (Devlin et al., 2019) on linguistically complex emotional text?

Firstly, we attempt to understand, in a strictly linguistic sense, the differences in the ways negative and positive emotions are manifested on social media. To investigate this we rely on the GoEmotions dataset (Demszky et al., 2020), which comprises English Reddit comments that have been manually annotated with 28 emotion categories. We investigate these texts' linguistic complexity by relying on diverse metrics from readability research. More specifically we consider a text more complex according to both lexical and syntactic complexity metrics.

In this first stage of our research, we investigate

possible correlations between these metrics and the polarity of the gold-standard emotion labels. In the second stage, we fine-tune three state-of-the-art transformers (BERT, RoBERTa, and SpanBERT) on the emotion dataset to perform multi-label classification.

We demonstrate that according to most metrics, negative emotions indeed tend to generate more complex text than positive emotions and that the transformer models are generally worse at predicting emotions for the more complex texts. We believe this calls for a more nuanced interpretation of the emotion detection performance of transformer models, with more detailed evaluations to have a better understanding of performance for complex data.

The remainder of this paper is organized as follows. Section 2 gives a brief overview of relevant related research, whereas Section 3 describes the data and information sources used for the experiments and analyses. Section 4 elaborates on the correlations between complexity and emotion polarity, whereas Section 5 discusses the machine learning experiments carried out to evaluate the performance of transformers on instances with various degrees of complexity. Section 6 ends this paper with concluding remarks and indications for future research.

## 2 Related Research

### 2.1 Negativity Bias and Complexity

The negativity bias or negativity effect is a cognitive bias that is described as the greater impact that negative stimuli have on humans compared to positive stimuli (Kanouse and Hanson Jr, 1987). This greater impact is not only reflected in observations that humans are more drawn to negative stimuli – which, for example, shows itself in more engagement with negative posts on social media (Rathje et al., 2021) or in higher consumption of negative news (Robertson et al., 2023) – but also in the fact that negative stimuli are associated with a higher cognitive load and more complex cognitive representations (Peeters and Czapinski, 1990). One could therefore ask whether this complexity is also reflected in language use.

Some researchers have indeed investigated the negativity bias in the context of language. An early but important observation is that there are more words for referring to negative emotional states than there are for referring to positive emotions (Averill, 1980). On a similar note, Jing-Schmidt et al. (2007) observed that there is a large number of emotive intensifiers that are based on negative emotion words, even to express positive messages (e.g., *"He is insanely good at his job."* or *"This was terribly sweet of you."*).

Besides focusing on the prevalence of negative vocabulary, few studies have dealt with examining the relationship between language complexity and negative emotions. Some studies have examined the language used in personal narratives about positive and negative experiences, but findings regarding complexity are inconclusive: in the work of Bohanek et al. (2005) on narratives of female undergraduates, it was found that positive narratives were more coherent and complex, although negative narratives were longer. However, in a study by Fivush et al. (2003), in which children were asked to narrate highly negative and highly positive events, it was found that narratives about negative events were more coherent. Moreover, these stories exhibited more emotion-related and cognition-related words, while the narrations about positive events focused more on descriptions of objects and people and were less coherent. These findings are more in line with the work of Guillory et al. (2011), in which emotion contagion in online groups was studied and it was found that negative emotions are correlated with more complex language. However, seeing these mixed findings, a thorough study using well-designed complexity metrics is desired to reliably examine the relationship between negative emotions and linguistic complexity.

### 2.2 Text Complexity

Text complexity can be studied from different angles. The first strand of experimental research investigates complexity by measuring processing effort when reading sentences. Effort can then be captured by cognitive proxies, such as for instance eye-tracking metrics (King and Just, 1991).

A second strand of research, more related to the work proposed here, is research on measuring text readability. Pioneering work in this respect has been performed by pedagogical researchers who designed traditional readability formulas, such as Flesch (1948) or Dale and Chall (1948). These formulas calculate text readability based on shallow surface text characteristics, such as the average word or sentence length or frequencies according to a predefined list. Thanks to advances in both

NLP and machine learning, we are currently able to derive more intricate text characteristics and consequently the research evolved towards readability prediction using both feature-based (Collins-Thompson, 2014; De Clercq and Hoste, 2016) and deep learning approaches, though the latter mostly rely on a combination with explicitly derived linguistic features (Meng et al., 2020; Lee et al., 2021). Interestingly, among the text characteristics that have been identified as most predictive when it comes to assessing a text's readability, the superficial ones – as employed in the traditional formulas – are often retained (François and Miltsakaki, 2012; De Clercq and Hoste, 2016). Additionally, syntactic features derived from parse tree information have also proven important indicators of a text's complexity as first revealed in the work by Schwarm and Ostendorf (2005) relying on a constituency parser and later also corroborated using dependency parsing information (De Clercq and Hoste, 2016). Considering the latter, many efforts have been devoted towards universal dependencies (De Marneffe et al., 2021), which was employed in recent research to operationalize how cognitive effort can be increased by complex syntax during translation through a variety of syntactic complexity metrics (Zou et al., 2022).

### 2.3 Emotion Detection

Given the abundance of opinionated data on the web, there is significant interest in methods to automatically analyze the emotional information within this data, e.g. in the context of market analysis and customer satisfaction for business intelligence (Song et al., 2021) or reputation management (Mohammad et al., 2015).

While initially, the aim was to identify the semantic polarity (positive, negative, or neutral) of a given instance, the field evolved towards a more fine-grained paradigm where the goal is to identify specific emotions instead of mere polarity orientations.

In line with this interest in fine-grained emotion detection, datasets labeled with a wide range of emotion categories have been developed. One of the largest and most fine-grained emotion datasets is GoEmotions (Demszky et al., 2020), which consists of English Reddit comments that were manually labeled with 28 emotion categories. As stated by the authors, it is "the largest human-annotated [emotion] dataset, with multiple annotations per

example for quality assurance". The applied emotion taxonomy includes 12 positive, 11 negative, 4 ambiguous emotion categories, and 1 "neutral" category.

Demszky et al. (2020) employed a BERT-based model and achieved an average F1-score of .46 on the fine-grained emotion classification task. The best performance was obtained for the positive emotions *gratitude* (.86), *amusement* (.80), and *love* (.78), while the model performed worst on *grief* (0), *relief* (.15) and *realization* (.21). Indeed, overall the performance was better on positive emotions (average F1-score of 0.52) compared to negative emotions (average F1-score of 0.41). The authors explain this by the presence of overt lexical markers for the best-predicted emotions, but also by frequency: overall, there are more instances with a positive emotion label than a negative label in the dataset. In this paper, we will therefore investigate whether complexity, besides frequency, could be an explaining factor in emotion detection performance.

## 3 Extracting Complexity Metrics

To investigate the link between emotions and complexity, we extracted a set of linguistic text characteristics that have shown to be good predictors for complexity in readability research (see Section 2.2). In the remainder of this paper, we will refer to these features as complexity metrics. In this section, we will first describe the datasets used for the correlation analysis and emotion detection experiments, and subsequently present an overview of all metrics that were extracted to measure complexity.

### 3.1 Datasets

In order to investigate the link between emotions and complexity on social media posts, we deliberately chose to work with a high-quality benchmark dataset for emotion detection that has been manually annotated.

The **GoEmotions dataset**[1] consists of English Reddit comments that were manually labeled with 28 emotion categories (including 12 positive, 11 negative, 4 ambiguous emotion categories, and 1 "neutral" category, allowing one to distinguish between subtle emotion expressions). The dataset consists of 58,009 instances. To create the Reddit corpus, first NLTK's word tokenizer was applied

---

[1]https://ai.googleblog.com/2021/10/goemotions-dataset-for-fine-grained.html

and only comments consisting of 3 to 30 tokens were selected. Subsequently, downsampling was performed to create a relatively balanced distribution of comments with varying lengths. We randomly selected a subset of 50,000 instances from a total of 58,009 instances in the simplified version of the dataset for automatic annotation of complexity information. In this evaluation set, the negative and positive polarities are fairly evenly distributed with 11,935 negative samples and 14,190 positive samples, while there are 23,875 neutral samples. In the fine-tuning experiments for Section 5, we use the entire dataset of 58,009 instances with an 80:10:10 split for training, validation, and testing respectively.

## 3.2 Complexity Metrics

To measure text complexity, we extracted a set of linguistic features that range from well-known superficial word shapes and lexical features to more complex syntactic features.

First, a set of word shape and lexical diversity and frequency features were extracted based on the readability research of De Clercq and Hoste (2016). We refer to this set as the **lexical** metrics. It comprises two more traditional readability features, namely *average word length* and *percentage of polysyllable words*, which have proven successful in readability work (François and Miltsakaki, 2012). Next, lexical complexity was modeled by measuring the *type-token ratio* and the percentage of words that can be found in the 1995 Dale and Chall list (Chall and Dale, 1995) for English. In addition, two-term weighting features were implemented that are often used to distinguish specialized terms in the text, namely *Term Frequency-Inverse Document Frequency*, aka tf-idf (Salton and Buckley, 1988) and the *Log-Likelihood* ratio (Rayson and Garside, 2000). For both of these features, the average values for all words in the text were calculated. Because connectives serve as an important indication of textual cohesion in a text (Graesser et al., 2004), we also counted the average number of connectives within the text. Finally, as named entity information provides us with a good estimation of the amount of world knowledge required to read and understand a particular text (De Clercq and Hoste, 2016), the number of unique entities was calculated as well.

Second, the following **syntactic** metrics were calculated based on the work of Zou et al. (2022), these metrics can be viewed as proxies to measure the impact of syntactic complexity on cognitive effort:

- IDT – Incomplete Dependency Theory:
  For a given token $t_i$, the IDT metric counts the number of incomplete dependencies between $t_i$ and $t_{i+1}$.

- DLT – Dependency Locality Theory:
  For a head token $t_i$, the DLT metric counts the number of discourse referents (proper nouns, nouns, and verbs) starting from $t_i$ ending to its longest leftmost dependent. The boundary-ending words should also be counted if they are discourse referents. For a non-head token, DLT is defined as zero.

- NND – Nested Noun Distance:
  The distance between two tokens $t_i$ and $t_j$ is their absolute positional distance, $|j - i|$. A noun $t_i$ is nested in another noun $t_j$ if $t_j$ is the ancestor of $t_i$ in the dependency tree. In any tree structure, a node that is connected to some lower-level nodes is called an ancestor. NND is the aggregation (i.e., sum, max or average) of nested noun distances. Following the research of Zou et al. (2022) we used *sum* as the aggregating metric.

- LE – Left-Embeddedness:
  The LE metric counts the number of tokens on the left-hand side of the main verb which are not verbs.

We have made the 50k subset of the GoEmotions dataset, automatically annotated with all 12 complexity metrics, available[2].

## 4 Correlation between Linguistic Complexity and Negative Emotions

In this section, we investigate whether there is a positive correlation between linguistic complexity and the presence of negative emotions. As explained in Section 3, we rely on syntactic and lexical readability metrics to measure complexity, while dividing the 28 emotion labels into 3 sets: positive (12 labels), negative (11 labels), and neutral and ambiguous (5 labels). As a preliminary

---

[2]https://huggingface.co/datasets/pranaydeeps/CAMEO

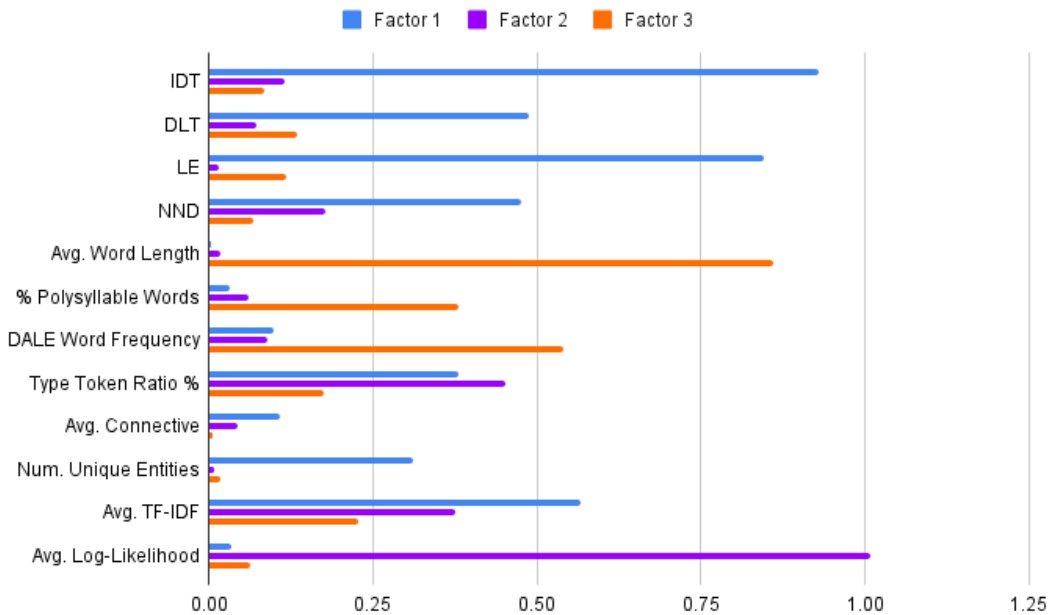

Figure 1: Factor loading matrix showing the importance of the considered readability metrics for three Factors.

attempt to study these correlations, we first perform an Exploratory Factor Analysis (EFA) to justify the use and selection of the complexity metrics before proceeding to investigate links between complexity and emotions. Moreover, by classifying the metrics into factors, we can further analyze the difficulties of transformers in modelling certain types of complexity.

First, we perform Barlett's Test of Sphericity (Armstrong and Soelberg, 1968) to confirm the validity of the factor analysis. The p-value was found to be 0.0 which means the observed correlation matrix is not an identity matrix and the test is statistically significant. We also perform a Keyser-Meyer-Olkin (KMO) Test which measures the proportion of variance among all the observed variables. The KMO for our data was found to be 0.659. A value higher than 0.6 is usually considered adequate for factor analysis.

To determine the adequate number of factors for decomposition, we set up a Scree plot, which is shown in Figure 2. Since 3 factors seem to have an eigenvalue higher than 1.0, we perform the factor analysis with 3 components. Figure 1 visualizes the factor loading matrix, demonstrating that Factor 1 largely combines syntactic metrics, while Factor 2 focuses on frequency-based metrics like TF-IDF, Log-Likelihood, and Type Token Ratio and Factor 3 solely focuses on basic lexical metrics like word length and polysyllable occurrences. The factor

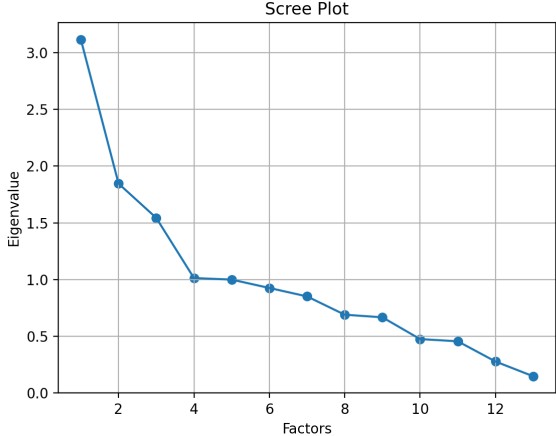

Figure 2: Scree plot showing the Eigenvalue in correspondence to the number of Factors.

variance reveals a cumulative value of 0.413, which means that a cumulative variance of 41.3% of the data is explained by the 3 factors. In Section 5 we further use the distinction between the factors to analyze the difficulties of transformer models in modeling certain types of linguistic complexity.

Since the metrics used seem to adequately represent linguistic complexity as statistically proven by the EFA, we can use these metrics to further understand the relationship between complexity and emotion polarities. To this end, we divide the complexity-enriched GoEmotions dataset into

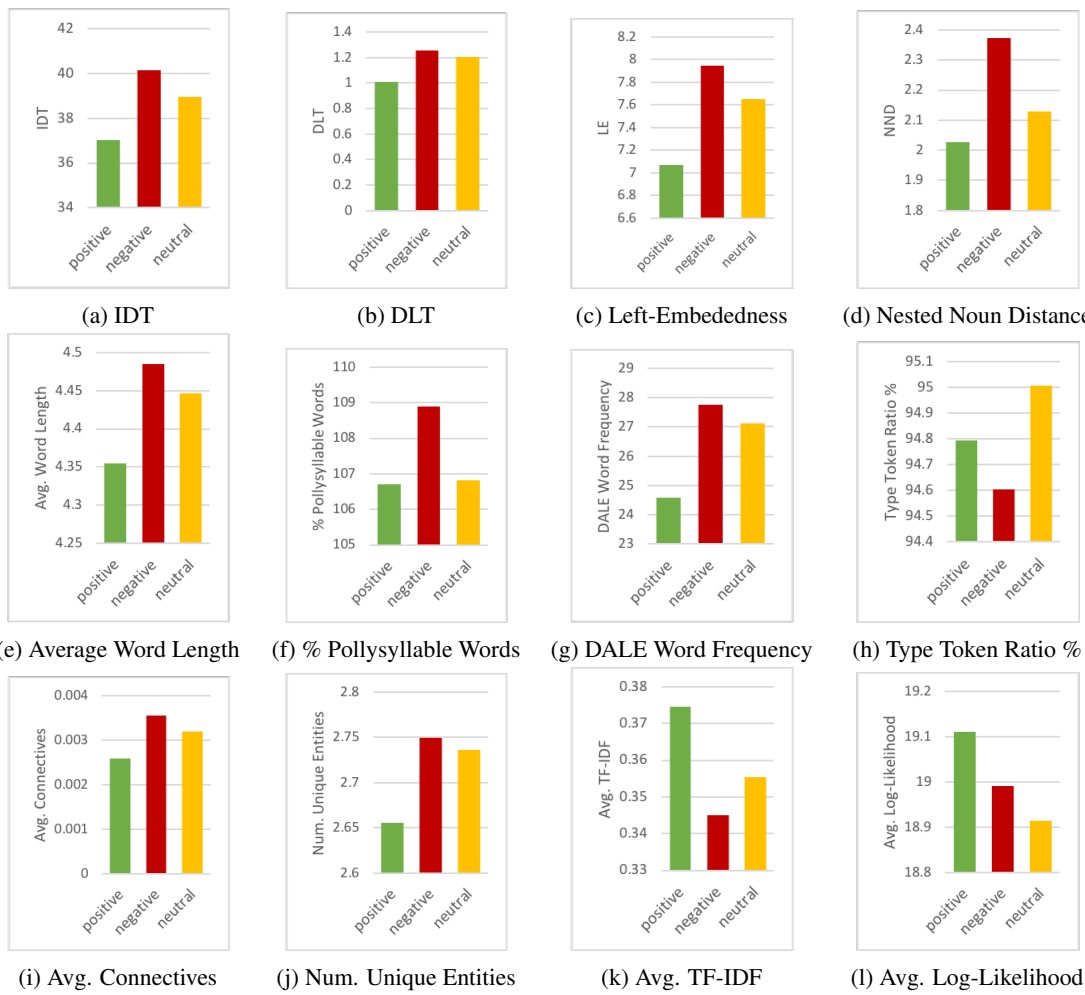

| (a) IDT | (b) DLT | (c) Left-Embededness | (d) Nested Noun Distance |
|---|---|---|---|
| (e) Average Word Length | (f) % Pollysyllable Words | (g) DALE Word Frequency | (h) Type Token Ratio % |
| (i) Avg. Connectives | (j) Num. Unique Entities | (k) Avg. TF-IDF | (l) Avg. Log-Likelihood |

Figure 3: Average value of each complexity metric for the broader emotion categories of positive, negative, and neutral for the GoEmotions evaluation set.

three polarities and calculate the average value for each metric for the three polarities individually. Figure 3 displays the averages for all 12 metrics for the three polarities. According to the figure, 9 out of the 12 metrics indeed have a higher average score for negative polarity. For some metrics, like Left-Embededness, Nested Noun Distance, or DALE Word Frequency, the difference between positive polarity averages is strikingly lower than the negative and neutral polarity averages. While Avg. TF-IDF, Avg. Log-Likelihood and Type Token Ratio % are anomalies, all three of these metrics convey correlated information represented by Factor 2 in the EFA (as illustrated in Figure 1).

## 5 Emotion Detection Evaluation

In this section, we attempt to understand how different state-of-the-art transformers are affected by complexity when classifying emotions using the GoEmotions datasets, which constitutes with its 28

emotion categories a very fine-grained classification task. For these experiments, we investigate 3 major transformers, BERT (Devlin et al., 2019), SpanBERT (Joshi et al., 2019), and RoBERTa (Liu et al., 2019). We fine-tune each of these models for multi-label classification for 20 epochs, with an initial learning rate of $2e-5$ and a weight decay of $0.1$, on a train set of 46,407 samples (80%) from the raw version of the GoEmotions dataset. We then use the best-performing checkpoint on a validation set of about 5,800 samples (10%). As an evaluation set, we use the remaining 5,802 samples (10%). All scores reported are averaged over 3 runs with different seeds.

To evaluate the output of the three fine-tuned transformer models, we again incorporate the metrics described in Section 3.2. We divide metrics using the EFA described in Section 4 into the 3 Factors, each representing a different aspect of linguistic complexity, i.e., syntactic complexity

|  |  | Factor 1 | | | Factor 2 | | | Factor 3 | | |
|---|---|---|---|---|---|---|---|---|---|---|
|  |  | Sim. | Med. | Comp. | Sim. | Med. | Comp. | Sim. | Med. | Comp. |
| Negative | | 275 | 889 | 214 | 29 | 1280 | 69 | 196 | 929 | 253 |
| Neutral | | 656 | 1733 | 381 | 46 | 2480 | 244 | 420 | 1825 | 525 |
| Positive | | 432 | 1023 | 199 | 34 | 1526 | 94 | 210 | 1138 | 236 |

Table 1: Distribution of Complex, Medium and Simple samples in the evaluation set considering the three main factors resulting from the Exploratory Factor Analysis (Section 4).

|  |  | Factor 1 | | | Factor 2 | | | Factor 3 | | |
|---|---|---|---|---|---|---|---|---|---|---|
|  |  | Sim. | Med. | Comp. | Sim. | Med. | Comp. | Sim. | Med. | Comp. |
| BERT | Negative | 0.2195 | 0.2068 | 0.2660 | 0.1034 | 0.2202 | 0.2480 | 0.2486 | 0.2053 | 0.2484 |
|  | Neutral | 0.4875 | 0.4331 | 0.3874 | 0.3544 | 0.4283 | 0.5741 | 0.4656 | 0.4262 | 0.4661 |
|  | Positive | 0.4850 | 0.4151 | 0.3665 | 0.3157 | 0.4312 | 0.4100 | 0.4733 | 0.4110 | 0.4540 |
| SpanBERT | Negative | 0.1148 | 0.1164 | 0.0732 | 0.0444 | 0.1097 | 0.1272 | 0.1346 | 0.1020 | 0.1173 |
|  | Neutral | 0.5176 | 0.4209 | 0.3793 | 0.2769 | 0.4278 | 0.5751 | 0.4574 | 0.4243 | 0.4736 |
|  | Positive | 0.4494 | 0.3848 | 0.3324 | 0.3235 | 0.3996 | 0.3516 | 0.4319 | 0.3802 | 0.4252 |
| RoBERTa | Negative | 0.3053 | 0.2508 | 0.2564 | 0.3157 | 0.2610 | 0.2647 | 0.3077 | 0.2500 | 0.2738 |
|  | Neutral | 0.4642 | 0.4275 | 0.3921 | 0.3000 | 0.4211 | 0.5658 | 0.4423 | 0.4199 | 0.4632 |
|  | Positive | 0.5027 | 0.4142 | 0.3644 | 0.3846 | 0.4352 | 0.3814 | 0.4774 | 0.4212 | 0.4248 |

Table 2: Performance (micro-F1) of the three transformer models fine-tuned for emotion detection with respect to the three main factors resulting from the EFA.

(Factor 1), metrics based on frequencies in a background corpus (Factor 2), and simple lexical complexity (Factor 3). For each set, if more than half of the metrics are in the top 20 percentile, we label the sample as a complex one (Comp.). If more than half of the metrics are in the bottom 20 percentile, we label the sample as a non-complex one (Sim.). The rest of the samples are labeled as medium complex (Med.). An overview of the distribution of the evaluation set according to this complexity labeling scheme is presented in Table 1.

The detailed emotion classification results, which are summarized in Table 2, provide some interesting findings. First and foremost, it is important to draw attention to the performance of the negative subclass of emotions which is consistently worse than the neutral or positive instances. This is proven by a two-tailed sample T-test, which proves the findings are statistically significant at $p < 0.05$. This is also apparent in Figure 4 where the performance for negative subsets for each factor is consistently poorer than for the positive subsets. The difference can be quite staggering: the Factor 1 Simple Positive set, for instance, has a micro F1-score of 0.4850 for the BERT model, while the Factor 1 Simple Negative set obtains a micro F1-score of 0.2195. This is in line with experimental

findings in psychology research that Negative Emotion Differentiation (ED) is harder than Positive ED for certain individuals (Starr et al., 2020), or otherwise put, that it is more difficult to distinguish between various fine-grained negative emotions than between different positive emotions, for humans and automatic systems alike.

A second noteworthy observation is that the Factor 1 Simple subsets tend to frequently outperform the Factor 1 Complex and Medium subsets, as illustrated by Figure 4(a). This is again verified by a two-tailed sample T-test, which proves statistically significant at $p < 0.05$. This is in accordance with the presumption that complex text samples require a higher degree of Natural Language Understanding (NLU), which the current generation of transformers is not (yet) capable of. This observation seems contrasted by the better performance on most Factor 2 Complex subsets, which often even surpasses the results obtained on the Factor 2 Simple subsets. This, however, might be linked to the fact that lexical complexity also comes with lexical richness, i.e., the presence of distinguishing keywords seldom used, which was one of the complexity metrics (type-token ratio) we used to categorize the evaluation set according to complexity, and therefore potentially a key factor in the decision-making for ED. Whereas, for Factor 3,

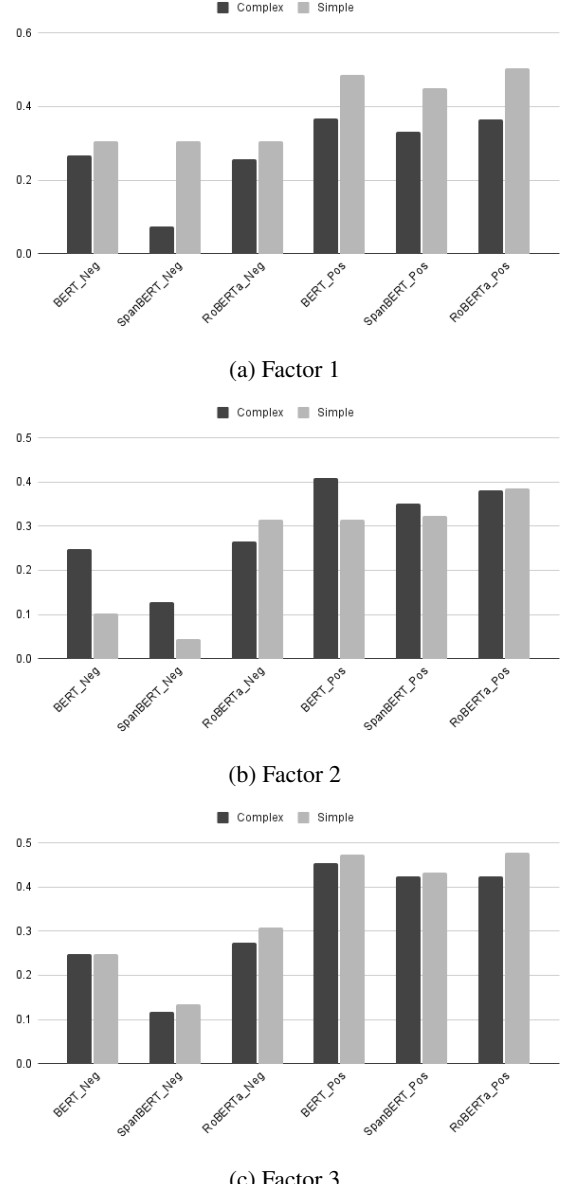

(a) Factor 1

(b) Factor 2

(c) Factor 3

Figure 4: Classification performance of BERT, Span-BERT and RoBERTa (in micro-F1) for Complex vs. Simple subsets for the three Factors (Figure (a), (b), (c)) for Negative and Positive emotions.

the difference in Simple vs. Complex subsets is inconclusive. This suggests that modeling lexically complex samples is not a pertinent issue of these sets of transformers.

The final, rather disappointing, observation is that SpanBERT, which is designed to better represent and predict spans of text, and thus assumed to better model more complex syntax, does not obtain better classification scores than BERT or RoBERTa on the complex subsets. In fact, SpanBERT only outperforms BERT and RoBERTa for some emotionally neutral subsets, which might

point to the pre-training data lacking emotional richness. SpanBERT was trained on the same datasets as BERT, but the data was restricted to samples exceeding 512 sub-tokens only from Book-Corpus and Wikipedia, potentially inhibiting the ability of the model to understand non-standard short-form social media communication.

## 6 Conclusion

In this work, we attempted to answer two research questions on the relationship between linguistic complexity and emotion detection. For this purpose, we relied on the GoEmotions dataset, which is a high-quality benchmark dataset for emotion detection that has been manually annotated with 28 emotion categories.

In the first part of this research, we investigated whether negative emotions are more linguistically complex than positive ones. With the help of twelve complexity metrics sub-divided into three factors, we discovered that negative emotions indeed have a higher average degree of complexity than positive or neutral emotions. Secondly, we attempted to dive deeper into the evaluation of fine-tuned transformers for emotion detection, by dividing our evaluation set from the GoEmotions dataset into complex, medium, and simple samples. Through this nuanced evaluation of three state-of-the-art transformers (BERT, SpanBERT and RoBERTa), we find that syntactically complex samples are often misclassified. We also notice that irrespective of difficulty, all models tend to perform poorly on negative emotions. Finally, we also discover that even though SpanBERT encodes a higher degree of complexity, this does not extend to emotion detection of complex text, as it either performs worse or is comparable to transformers trained with standard masking (BERT and RoBERTa).

To conclude, we aim to encourage a more detailed evaluation of current emotion detection systems, by extending the evaluation setup to better understand the performance for more linguistically complex instances, rather than the reporting of aggregated scores which may only paint a half-formed picture. In future work, we would like to demonstrate the validity of the results on datasets from other domains and annotation strategies, as well as languages other than English. Further, since linguistic complexity does have an impact on emotion detection performance, we would like to investigate more intricate approaches to model more

complex samples.

## Limitations

As a first and primary limitation, we want to point out that the present study has only been validated on an English dataset, whereas emotions are global. Thus, more work needs to be done before the conclusions can be extended to other languages and domains. The study was also conducted using the GoEmotions dataset. Even though it can be considered the largest human-annotated emotion dataset in English, the results should be validated on more datasets and source domains. Lastly, the inferences in this work assume that the selected metrics are exhaustive and adequately represent the complexity of a text sample, however, assessing linguistic complexity is a complex and ongoing multidisciplinary area of research, and the selected metrics may be missing some information to properly distinguish complexity.

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
