# OpenReview forum: "Misery Loves Complexity: Exploring Linguistic Complexity in the Context of Emotion Detection"
_EMNLP/2023/Conference — EMNLP 2023 Findings_

### Official Review · Reviewer_hKnV · 2023-08-01

**Soundness:** 3

**Excitement:**

2: Mediocre: This paper makes marginal contributions (vs non-contemporaneous work), so I would rather not see it in the conference.

**Paper Topic And Main Contributions:**

This paper examines the relationship between linguistic complexity and the detection of negative emotions. The primary insights from this study are:

1. Texts expressing negative emotions tend to have a higher degree of complexity compared to those conveying positive or neutral emotions.
2. Transformer-based models are more likely to misclassify texts that are syntactically complex and those expressing negative emotions.

**Questions For The Authors:**

A. Can you provide insights or explanations as to why TF-IDF, Avg. Log-Likelihood, and Type Token Ratio % exhibit higher average scores for positive polarity? Does this suggest that positive emotions have greater lexical complexity, given that all three of these metrics are lexical metrics?

B. Regarding the 120,000 samples from the raw dataset version used to train the classifiers, do they ensure a balanced representation of both positive and negative samples, as well as an even distribution between complex and simple samples?

**Reasons To Accept:**

This study provides insights for enhancing existing emotion detection models. It raises the issue that text expressing negative emotions and those with syntactic complexity pose greater challenges for model classification, and thus potentially need to be addressed specifically.

**Reasons To Reject:**

The paper's limitations are significant:

1. The correlation analysis is preliminary. Conclusions are made solely based on average scores of metrics for positive and negative samples. Comprehensive testing, including significance tests, is needed.
2. The emotion detection evaluation lacks depth. Only basic experiments and metrics (like F1 score) were performed and reported.
3. State-of-the-art emotion and sentiment detection models should also be assessed.
4. There's a notable absence of detailed analysis explaining anomalies in certain metrics when assessing correlations between negative and positive emotions (refer to Question A).

**Reproducibility:**

4: Could mostly reproduce the results, but there may be some variation because of sample variance or minor variations in their interpretation of the protocol or method.

**Reviewer Confidence:**

4: Quite sure. I tried to check the important points carefully. It's unlikely, though conceivable, that I missed something that should affect my ratings.

---

> ### Author Rebuttal · Authors · 2023-08-29
>
> Respected Reviewer,
>
>
> Thank you for your feedback and extremely constructive comments for our work. First of all, we would like to address the question put forth to us:
>
> A. Given that the majority (5 out of the other 8) of lexical complexity metrics point to the opposite, we would still say that negative emotions have a higher lexical complexity. However, it is an interesting line of discussion as to why TF-IDF, Avg. LL and TTR% have a higher average for positive emotions. It is also interesting to note that all three of these features belong to Factor 2 in the EFA and might encode very similar information. For TF-IDF and Avg. LL, the scores could largely be influenced by the frequencies in the background corpora used for calculating these metrics as well.
>
> B. The 120,000 or so samples used for training do indeed have a near identical distribution of positive-negative samples, however the complex/simple distribution cannot be verified because only the evaluation set of 50,000 was annotated with complexity metrics, since some of the complexity metrics are extremely time intensive to calculate and the task would be quite expensive computationally. However, we agree that it would be fairer to use the same set of 50,000 examples for the finetuning experiments (in a cross-validation setting), so that we have complexity metrics for all used and evaluated instances. Therefore, we have already started the finetuning experiments in a 5-fold setting on this smaller dataset. We will report these results in the camera-ready version upon acceptance.
>
>
> Additionally, we would also like to address some of the criticisms posed:
>
>
> 1. Based on the suggestion of other reviewers we agree that the correlation analysis and the emotion detection evaluation can be improved by taking into account the different factors produced during the EFA and analyzing the factors independently, pointing out anomalies, and providing more examples. Significance tests, therefore, could also be an excellent improvement for this section.
>
> 2. For now, we have opted to only evaluate BERT, RoBERTa, and SpanBERT. This was a conscious choice and we wanted to focus the evaluation on the comparisons between SpanBERT (designed for higher complexity) versus BERT and RoBERTa (designed for general use cases). However, we do plan to continue this line of research and conduct similar studies on the zero-shot, few-shot emotion detection capabilities of generative models.

---

### Official Review · Reviewer_nrYf · 2023-08-05

**Soundness:** 3

**Excitement:**

3: Ambivalent: It has merits (e.g., it reports state-of-the-art results, the idea is nice), but there are key weaknesses (e.g., it describes incremental work), and it can significantly benefit from another round of revision. However, I won't object to accepting it if my co-reviewers champion it.

**Missing References:**

None

**Paper Topic And Main Contributions:**

This paper studies negativity bias in social media posts, specifically by asking (i) whether posts conveying negative emotions are more linguistically complex than positive ones and (ii) whether a text’s linguistic complexity increases the difficulty to model the related emotions using Transformers. To do so, the paper analyses the GoEmotions dataset, a collection of over 200,000 Reddit posts manually annotated according to 28 emotion categories (12 positive, 11 negative, 4 ambiguous, and 1 neutral).
For the first research question, the authors downsampled from that dataset to obtain a balanced set of 50,000 examples. Linguistic complexity is modeled via lexical metrics (e.g., average word length and percentage of polysyllable words, TF-IDF) and syntactic metrics (based on dependency-tree indicators). Results show that out of the 12 metrics used, 9 show a higher score for negative emotions as compared to positive ones.
For the second one, the authors experiment with the full 200,000 examples dataset, and fine-tune three Transformer models (BERT, SpanBERT, and RoBERTa). To do so, the authors utilize the aforementioned complexity metrics and divide dataset samples into simple, medium, and complex. The authors notice first that classification performance is lower for negative emotions than for positive ones. Furthermore, the simple instances seem to be more easily correctly classified by the models as compared to the medium and complex ones.

**Questions For The Authors:**

Did you experiment with different hyperparameters for fine-tuning the individual baseline models? This could potentially help in further understanding and explaining the performance differences between them.

**Reasons To Accept:**

* The paper provides a detailed introduction to the psychological related works motivating their research objective.
* The complexity analysis relies on a wide range of features, which speaks to the robustness of the obtained results.
* Overall, the paper is well-written, and the analysis is technically sound.


**Reasons To Reject:**

* The paper describes the task as “explor[ing] the idea of a negativity bias from the perspective of the author” (lines 51-52). However, in the context of the GoEmotions dataset, it is important to make the distinction between the emotions experienced by the authors and those identified by the annotators. Since in the GoEmotions dataset texts have not been annotated by the authors themselves, relying on this dataset has its limitations (which should be addressed in the paper).
* Given recent advancements in large language models, I would be curious to see how zero-shot and few-shot learning with larger models perform as compared to the BERT-based models.


**Reproducibility:**

3: Could reproduce the results with some difficulty. The settings of parameters are underspecified or subjectively determined; the training/evaluation data are not widely available.

**Reviewer Confidence:**

3: Pretty sure, but there's a chance I missed something. Although I have a good feel for this area in general, I did not carefully check the paper's details, e.g., the math, experimental design, or novelty.

**Typos Grammar Style And Presentation Improvements:**

Table 2: I’d recommend mentioning the performance metric (F1) being used here to avoid confusion.

---

> ### Author Rebuttal · Authors · 2023-08-29
>
> Respected Reviewer,
>
>
> Thank you for your feedback and extremely constructive comments on our work. First of all, we would like to address the question put forth to us:
>
>
> A. Yes, we optimized the hyper-parameters of each of the baseline models individually and implemented early stopping to ensure we had the best checkpoint for each transformer.
>
>
> Additionally, we also want to address some of the criticisms pointed out:
>
>
> 1. We indeed acknowledge that the emotion annotations have not been made by the authors themselves and can thus only be considered as a proxy. We do know that the instructions for the annotators were to identify the emotions as expressed by the writer of the text and that good inter-rater agreement was reported. If our paper gets accepted we will definitely add this as one of the limitations of our work.
>
> 2. We consciously opted to only evaluate transformers with similar architectures and especially the difference between models like BERT & RoBERTa, designed for generic use versus SpanBERT, designed to understand higher complexity. However, in future work, we would indeed like to explore how the next generation of auto-regressive models deal with the complexity of emotions.
>
> 3. We thank you for pointing out the suggestion for Table 2, we will be sure to update it.

---

### Official Review · Reviewer_z7G8 · 2023-08-05

**Soundness:** 3

**Excitement:**

4: Strong: This paper deepens the understanding of some phenomenon or lowers the barriers to an existing research direction.

**Paper Topic And Main Contributions:**

This paper explores the connection between language complexity and emotions, and specifically examines how these factors impact the performance of transformer-based emotion detection models. The findings reveal that negative emotions tend to produce more complex text compared to positive emotions, and transformer models generally struggle more with complex language.

The main contributions of this research are twofold. First, the study addresses intriguing research questions inspired by previous work in linguistics. By providing empirical evidence through quantitative results, the paper offers valuable insights that support and validate existing theories. Second, the authors present a well-designed set of experiments that incorporate various metrics to capture different aspects of language complexity. This comprehensive approach enhances the understanding of the relationship between language complexity and emotions.


**Questions For The Authors:**

A. What are the main differences between the raw and simplified versions of the GoEmotions dataset (L2787-279)? Additionally, why was a subset of 50k instances randomly selected in the simplified version (L285), while the raw version was used for the fine-tuning experiments (L292-293)?

B. Have you considered including other available datasets for emotion detection, such as HurricaneEmo?

C. What is the primary motivation for conducting the factor analysis in Section 4? Initially, I initially thought it was to categorize the metrics into three categories (syntactic metrics, frequency-based metrics, lexical metrics) for subsequent analyses. However, the paper appears to have used either the entire set of features (Figure 3) or the two categories (Table 1 and 2), leaving it unclear what additional insights the factor analysis contributes to the paper other than showing there are three groups of features in the analyses.

D. What was the motivation behind your definition of "complex" and "non-complex" examples discussed in lines 451-453? How was the specific criteria (i.e., top/bottom 20 percentile for more than half of the metrics) determined?

E. Why do you think the metrics represented by Factor 2 were outliers in Figure 3? Do you have any good explanations?


**Reasons To Accept:**

- The research questions are interesting, novel, and well-motivated by prior research.
- The methods and experiments are well-thought-out.
- The paper is well-written, easy to read, and clearly presents their findings and contributions.


**Reasons To Reject:**

- Limited generalizability: All experiments were conducted solely on one data set, GoEmotions, which may limit the broader applicability of the findings.
- Some parts of the paper were slightly confusing. For instance, the motivation behind conducting factor analysis over the complexity metrics is not clear. Additionally, the reasoning for not grouping the features into three groups based on the factor analysis in the later model performance analyses, and instead using binary syntactic and lexical feature categorization, could use some explanation.
- The method, findings, and contributions of the paper might be better suited for a shorter publication rather than a long paper. More in-depth and fine-grained analyses of the results, such as exploring the specific factors contributing to the complexity of negative language with examples and differences among the fine-grained negative emotions, could have made the paper stronger. However, the current draft does not provide much detail in this regard.


**Reproducibility:**

4: Could mostly reproduce the results, but there may be some variation because of sample variance or minor variations in their interpretation of the protocol or method.

**Reviewer Confidence:**

4: Quite sure. I tried to check the important points carefully. It's unlikely, though conceivable, that I missed something that should affect my ratings.

**Typos Grammar Style And Presentation Improvements:**

- The design, image quality, readability, and legends could be improved for all figures in the paper.
- Tables 1 and 2 contain an excessive amount of numbers, making them challenging to read. Consider converting Table 1 into a stacked bar graph, and use different number formatting (e.g., 0.4409 → 44.1) for Table 2 to enhance clarity.

---

> ### Author Rebuttal · Authors · 2023-08-29
>
> Respected Reviewer,
>
>
> Thank you for your feedback and extremely constructive comments on our work. First of all, we would like to address the questions put forth to us:
>
>
> A. The raw and simplified versions of the data are taken directly from the official versions uploaded on the huggingface datasets hub here: https://huggingface.co/datasets/go_emotions. However, it is unclear from the paper and GitHub repository how the authors selected the reduced 58,009 instances for the simplified version. According to the dataset card: “In the raw data, labels are listed as their own columns with binary 0/1 entries rather than a list of [label] IDs as in the simplified data”. Additionally, the raw version consists of additional meta-data which is left out in the simplified version. Finally, the simplified version is divided into pre-existing train/test/val splits.
>
> B. We have considered using additional datasets to validate our hypothesis, however, at the current stage, we could not identify any other datasets that fulfilled all our requirements, since we wanted to analyze only user-generated content, which was manually annotated, with fine-grained emotions and not the standard coarse Ekman or Plutchik labels. Therefore, datasets like Dair-AI and HurricaneEmo did not fulfill our requirements.
>
> C. The factor analysis was conducted to determine the validity of the selected complexity metrics, to check for information overlap among the metrics, and as pointed out, to demonstrate that the metrics largely belong to 3 broad categories. We indeed really like and appreciate the suggestion of also using these broad factor categories for further analysis in the later section and we will implement this in the updated version of the paper.
>
> D. This was an intuitive decision: taking the top/bottom 20% ensures that we have a high chance of really capturing complex and non-complex instances. However, being in the top/bottom 20% was not required for all metrics, but for at least half of them, because tightening the requirement to all metrics would leave us with too little data to analyze.
>
> E. We hypothesize that TF-IDF and Avg. LL are likely outliers due to a large influence of frequencies in the background corpora used to calculate these metrics.
>
>
> Additionally, according to suggestions, we would like to make the following changes to the paper:
>
> 1. Improve the quality of figures and tables, and make suggested changes to Table 1.
> 2. Add more detailed examples, explore the factors contributing to the complexity, discuss anomalies, etc.

---

### Meta-Review · Area_Chair_87mG · 2023-09-20

**Recommendation:** 4

**Metareview:**

The paper explores the relation between language complexity when expressing emotions, and the performance of transformer-based models on emotion classificaition. The experimental setul and analysis are sound as highlighted by all reviewers. Reviewers found the paper moderately exciting (i.e. interesting research question grounded on linguistic theory).

---

### Decision · Program_Chairs · 2023-10-07

**Decision:**

Accept-Findings

**Comment:**

The paper explores the relation between language complexity when expressing emotions, and the performance of transformer-based models on emotion classificaition. The experimental setul and analysis are sound as highlighted by all reviewers. Reviewers found the paper moderately exciting (i.e. interesting research question grounded on linguistic theory).